# Uncovering the Role of PdePrx12 Peroxidase in Enhancing Disease Resistance in Poplar Trees

**DOI:** 10.3390/jof9040410

**Published:** 2023-03-27

**Authors:** Guanghua Cai, Yan Zhang, Liyu Huang, Nian Wang

**Affiliations:** 1College of Horticulture and Forestry Sciences, Huazhong Agricultural University, Wuhan 430070, China; 2Hubei Engineering Technology Research Center for Forestry Information, Huazhong Agricultural University, Wuhan 430070, China

**Keywords:** *PdePrx12* gene, Class III peroxidase, disease resistance, NL895 poplar

## Abstract

Peroxidase (Prx)-related genes are reported to be involved in the metabolism of hydrogen peroxide (H_2_O_2_) in plants. Here, we found that the expression of the *PdePrx12* gene was upregulated in wild-type (WT) poplar line NL895 infected with the pathogens *Botryosphaeria dothidea* strain 3C and *Alternaria alternata* strain 3E. The *PdePrx12* gene was cloned in the poplar line NL895 and its overexpression (OE) and reduced-expression (RE) vectors were constructed. OE and RE transgenic lines were then generated. The H_2_O_2_ content in the leaves was measured by DAB staining and spectrophotometric analysis, and the data revealed that the OE line had a reduced H_2_O_2_ content, whereas the RE line had an increased H_2_O_2_ content. These transgenic and WT plants were also inoculated with the 3C/3E pathogens. The leaf area infected by pathogen 3C/3E was determined and the OE line was found to have a larger area of infection, whereas the RE line was found to have a smaller area of infection. This result suggested *PdePRX12* is involved in disease resistance in poplar. Given these results, this study demonstrated that when poplar is infected by pathogens, the expression of *PdePrx12* is inhibited, leading to an increase in H_2_O_2_ content, thereby enhancing disease resistance.

## 1. Introduction

The appropriate amount of reactive oxygen species (ROS) produced by cells can act as a regulator of signal transduction pathways. When exposed to stress, excessive intracellular production of ROS exceeds the capacity of the reducing buffer system and can cause damage to cells through membrane peroxidation and delipidation, and it can even cause cell death [1]. In turn, cells synthesize antioxidant enzymes (such as superoxide dismutase (SOD), catalase (CAT), ascorbate peroxidase (APX), glutathione peroxidase (GPX), and the peroxidase (Prxs)) in order to protect themselves from oxidative damage [2]. Prxs are a class of peroxidases widely found in eukaryotes and prokaryotes, are ubiquitous in organisms, and act independently of other peroxidases [3], of which Class III peroxidases compose a multigene superfamily of antioxidant proteins unique to plants [4]. In higher plants, Prx-related genes function in many plant biological activities and are involved in regulating the metabolism of plant hormones, regulating the metabolism of plant signaling molecules (such as ROS, RNS, and so on), altering the length and thickness of cell walls, and participating in the process of disease resistance regulation in plants. In 1965, with the discovery of Prx involvement in the metabolic process of auxin, increasing numbers of studies have shown that Prxs can catalyze auxin oxidation and decarboxylation, and it has now been demonstrated that Prxs are involved in lignin production in tobacco, *Arabidopsis*, and several species of gymnosperms and angiosperms. Reverse genetics is a recognized method for inferring validated gene functions through the study of the individual phenotypes of specific target genes. Prx gene expression can cause observable phenotypes such as lignification, cell elongation, stress defense, and seed germination [5]. In recent years, Prx gene studies have been conducted, and some progress has been made in model plant species such as *Arabidopsis* and rice in order to better understand the role of Class III peroxidases in plant growth and development. However, the cloning and functional identification of Prx genes in poplar have been rarely reported.

ROS accumulate in a variety of plant disease defense systems, and ROS are important for pathogen defense and signal transduction. ROS not only kill pathogens directly, but also promote cell wall lignification and HR-related cell death. In addition, as secondary messengers, ROS can activate the expression of defense genes, ultimately leading to resistance in plants [6]. When plants are subjected to biotic or abiotic stresses, they are able to form similar physical structural barriers or produce ROS and RNS signaling molecules to form a highly toxic chemical region to inhibit the expansion of the pathogen infection area. ROS in plants mainly include O_2_·^−^, OH^−^, ·OH, H_2_O_2_, and ^1^O_2_· [7,8,9]. In general, a low concentration of H_2_O_2_ functions as a signal molecule, while high accumulation of H_2_O_2_ can cause toxicity in plants and thus cell death [10,11,12]. The study shows that H_2_O_2_ is generally not produced in cells with rapidly elongating cell walls but there is a strong release of H_2_O_2_ in injured cells or cells under stress mechanisms [13]. Doke first reported the production of H_2_O_2_ in potato leaves when the potato was infested with pathogens in 1983, and hypersensitive response occurred, thereby limiting the harm of pathogens to a certain location without spreading and minimizing the harm to plants [14,15]. Thus, H_2_O_2_ and even ROS play a key role in the plant disease resistance response.

Poplar is an important timber forest species in China, is planted in a wide area, and is largely used not only for its timber but also as a raw material for paper and house construction, but due to various natural and man-made reasons, poplar pests and diseases are very common. Early poplar plantations were dominated by pure stands, and this single-stand structure resulted in poplar trees being subjected to a variety of diseases [16]. These diseases mainly include leaf spot, ulcers, leaf rust, leaf blight, and branch and stem rot [17]. Poplar ulcer is the most common disease on branches and trunks, and the causal pathogen is *Botryosphaeria dothidea* [18]. Causing premature leaf abscission, poplar leaf blight is extremely harmful to poplar leaves, and its causal pathogen is *Alternaria alternata* [19].

Poplar has many pests and diseases, and traditional control measures not only pollute the environment but also can generate some resistance and waste human resources. Thus, the ideal solution is to select and breed disease-resistant varieties. Poplar trees grow rapidly and have a short breeding cycle. Because of years of effort, it is now easy to generate transgenic lines on poplar trees through the Agrobacterium-mediated genetic transformation method [20]. In this study, the expression of the *PdePrx12* gene was upregulated in wild-type (WT) poplar line NL895 infected with the pathogens *Botryosphaeria dothidea* strain 3C and *Alternaria alternata* strain 3E. The overexpression (OE) and reduced-expression (RE) of *PdePrx12* transgenic lines were generated. The H_2_O_2_ content in the leaves was measured and the data showed that OE line had a reduced H_2_O_2_ content, while the RE line had a reduced H_2_O_2_ content. These transgenic and WT plants were also inoculated with the 3C/3E pathogens, and the result demonstrated that the *PdePrx12* gene is involved in disease resistance in poplar. Overall, this study revealed that *PdePrx12*, which encodes a peroxidase, is involved in disease resistance in poplar.

## 2. Materials and Methods

### 2.1. Experimental Materials, Growth Environment and Treatment Conditions

The hybrid poplar (*Populus deltoids* × *Populus euramericana*) line NL895 was used as material for tissue culture in woody media (WPM) at 25–28 °C under a photoperiod of 16 h of light and 8 h of darkness and was passaged once a month. The subsequent culture conditions included the use of soil, 1/2 Hoagland’s solution and incubation in a growth chamber at a temperature of 21–25 °C with a photoperiod of 16 h of light and 8 h of darkness [21]. The poplar pathogen *B. dothidea* strain 3C and poplar leaf blight pathogen *A. alternata* strain 3E were inoculated in potato dextrose agar (PDA) media [22] and incubated in a 28 °C incubator protected from light for approximately 7 days for pathogen 3C and approximately 14 days for pathogen 3E. These pathogen strains were stored in our laboratory.

To verify the expression pattern of the *PdePrx12* gene after pathogen infection, the 4th-10th flattened leaves of WT plants cultivated in the soil for 2 months were selected, and the pathogen 3C/3E and PDA were cut into plots of the same size first and inoculated on the abaxial side of the leaves. Inoculation with pathogen 3C/3E constituted the experimental group, and inoculation with PDA constituted the control group. Samples inoculated with pathogen 3C were collected on Days 0, 3 and 7, and samples inoculated with pathogen 3E were collected on Days 0, 8 and 14, while the corresponding controls were collected. The material that included the leaf inoculation site (1 cm^2^) was quickly placed in liquid nitrogen and stored at −80 °C until use.

### 2.2. RNA Extraction with Reverse Transcription and Quantitative PCR (RT–qPCR) Analysis

Total plant RNA was extracted and analyzed by quantitative real-time PCR (qRT–PCR) according to our previous methods [23]. RNA was extracted from poplar tissues using a kit what named RNAprep Pure Plant Total RNA Extraction Kit (DP432) (Tiangen Biochemical Technology, Co, Shanghai, China). To determine the expression of *PdePrx12* in different tissues, tissue culture plants of NL895 with consistent growth status were selected, and root, stem, leaf, petiole and shoot tissues were collected for RNA extraction. To determine the expression of *PdePrx12* upon infected with pathogen 3C/3E, leaf samples (1.5 × 1.5 cm^2^ in size) surrounding the center of the disease spot were collected after removal of the clumps when the pathogen 3C/3E was allowed to infect for a certain number of days, and their RNA was extracted. Three biological replicates were used for all qRT-PCRs. Additionally, different biological replicates represented samples were collected from different plants. Actin (gene sequence number: *Potri.001G309500*) was used as the internal reference gene for qRT–PCR in this study. Quantitative primers were designed based on the reference genome of *P. trichocarpa*, and the primer sequences used are shown in Appendix A. Based on the qRT–PCR data, relative gene expression was calculated using the 2^−ΔΔCT^ method [24].

### 2.3. Generation and Identification of Transgenic Lines

The cDNA of WT NL895 poplar was used as a template to amplify the full-length coding DNA sequence of the *PdePrx12* gene and adding the 2× Hieff Canace^®^ PCR Master Mix (10136ES03, Yeasen Biotechnology, Co, Shanghai, China) into the system. The target gene was ligated into a 2301S vector by the homologous recombination method to construct an overexpression vector (OE-PdePrx12). A 266 bp fragment of the *PdePrx12* gene was amplified using specific primers containing the attb adaptor and ligated to a pHellsgate4 vector to construct a reduced expression vector (RE-PdePrx12). These vectors and their construction methods are referred to in our previous study [25]. The constructed OE and RE vectors were subsequently transferred into *Agrobacterium tumefaciens* strain GV3101 according to a previous poplar transgenic method, and *Agrobacterium tumefaciens*-mediated transformation of NL895 poplar was performed [26]. After obtaining transgenic plants, three levels of positive screening were performed for DNA, RNA, and protein. For positive detection of overexpression (OE) lines, at the DNA level, PCR verification was performed using a combination of the gene-specific primer PdePrx12-F and vector primer 2301S-testR. At the RNA level, RNA was extracted from DNA-positive transgenic lines, and qRT–PCR was performed after reverse transcription of cDNA. At the protein level, in reference to a previous study [27], transgenic confirmation was performed via GUS staining of young leaves of transgenic plants. For positive screening involving DNA and RNA analysis only for RE lines, at the DNA level, the left primers PHGRV35SF and PHGRVOCS from the pHellsgate4 vector and the right gene-specific primer attb-PdePrx12-R for *PdePrx12* were used for PCR validation. At the RNA level, extraction of RNA from transgenic lines that were positive for DNA validation was reverse transcribed to cDNA and then subjected to qRT–PCR. All the primers used for DNA level detection are shown in Appendix A.

### 2.4. Determination of H_2_O_2_ Content

To investigate the effect of the *PdePrx12* gene on the H_2_O_2_ content in NL895 poplar tissues, two methods were used, which are described in our previous study [25]. Sixty milligrams of fresh leaf tissue was collected from each line and ground in liquid nitrogen. Then, 2 mL of 1 M HClO_4_ (containing 5% PVP) was added, and the material was ground until homogenous. The sample was then centrifuged at 4 °C at 12,000× *g* for 10 min, after which the supernatant was removed and adjusted to a pH of 5.6 with 5 M K_2_CO_3_. The sample was centrifuged again at 4 °C, 12,000× *g* or 30 s, after which 50 μL of the supernatant was removed. One unit of ascorbate oxidase was added, after which the sample was incubated for 10 min at room temperature. Last, 870 μL of 0.1 M PBS, 20 μL of 165 mM DMAB, 50 μL of 1.4 mM MBTH, and 1 U horseradish peroxidase were added, and the sample was incubated for 5 min at room temperature. The absorbance of the sample was measured at 590 nm, and the content of H_2_O_2_ was calculated according to the formula derived from a standard curve.

Next, the H_2_O_2_ content in plants was detected using 3,3′-diaminobenzidine (DAB) staining. The leaf tissues were directly immersed in 1 mg/mL DAB staining solution and subjected to a vacuum for 30 min, after which the samples were incubated overnight at 37 °C and then placed in 95% ethanol in a 60 °C water bath to decolorize.

### 2.5. Phenotypic Observations of the Symptoms

The obtained positive transgenic lines and WT NL895 poplar lines were grown in tissue culture bottles whose tops were open and that were filled with water. After the transgenic lines were acclimated to the external environment (approximately 5 days), they were grown hydroponically for 20 days, after which they were planted in nutrient-enriched soil and watered regularly. After 60 days, they were used for pathogen inoculation. WT NL895 and transgenic lines that grew normally were selected, and the 4th–10th (counting from top to bottom) noncurled normal leaves were rinsed with water and placed in a 15 cm diameter plastic dish lined with 2 layers of moistened filter paper, the petioles were wrapped in a cotton ball, and the abaxial side the leaf faced up. A similar size of PDA plots (a circle with 0.5 cm in diameter) infected with the pathogen strain were prepared. Two pieces of PDA plots were placed symmetrically on the abaxial surface of the leaf on nonvein sites, after which the dishes were sealed. After inoculation, the dishes were placed in an environment at 28 °C and incubated in darkness. The expansion of the pathogen on the leaves was observed daily, water was added when appropriate, and when the pathogen developed to a significant difference, the clumps were removed, the leaves were scanned, and the absolute area of individual spot infection development was measured using Image-Pro Plus image processing software. All the spot data were then statistically analyzed.

### 2.6. Construction of a Phylogenetic Tree

The protein sequence coded by the *PdePrx12* gene was imported into the *Arabidopsis* database website (https://www.arabidopsis.org/, (accessed on 1 December 2022) and database is Araport11), and the protein sequences of the 10 genes with the highest paired values were extracted. The extracted protein sequences of the *Arabidopsis thaliana* gene, hairy poplar *Potri.005G195600,* and *PdePrx12* gene of NL895 poplar were imported into Mega X software together and compared by the ClustalW program. Then, a neighbor-joining tree was constructed, and the number of bootstrap replicates was set to 1000.

### 2.7. Domain of Protein

The amino acid sequence of PdePrx12 protein was imported into the protein domain website (http://pfam.xfam.org, (accessed on 1 December 2022)) to predict its protein domain.

## 3. Statistical Analysis

All statistical analyses were performed using R software downloaded (https://www.r-project.org/, (accessed on 1 December 2022)). The “aov” and “Tukey HSD” functions were used in R software for ANOVA and multiple comparisons, respectively.

## 4. Results

### 4.1. The PdePrx12 Gene Is Homologous to the AtPrx12 Gene

In a previous study, we investigated gene expression patterns in poplar via RNA-seq data [28]. The results indicated that the *Potri.005G195600* gene (*Populus trichocarpa* v3.0) responds to pathogen infection. A gene encoding a protein with high similarity to *AtPrx12* was identified and cloned. Due to its high similarity to *AtPrx12*, this gene was named *PdePrx12*. The primers PdePrx12-F and PdePrx12-R and attb-PdePrx12-F and attb-PdePrx12-R were designed based on the CDS of the *Potri.005G195600* gene of *P. trichocarpa* for the amplification of the *PdePrx12* gene. The gene has three exons and two intron regions (Figure 1A). The top 10 genes in *Arabidopsis* that were highly similar to the *PdePrx12* gene and the *Potri.005G195600* gene in *Populus trichocarpa* were selected for construction of a phylogenetic tree. The *PdePrx12* gene was classified into the same subbranch as the *Potri.005G195600* gene of *P. trichocarpa* and the *AtPrx12* gene (Figure 1B), which suggests that the *PdePrx12* gene in the NL895 poplar genome corresponds to the *AtPrx12* gene. The sequences of the *AtPrx12*, *Potri.005G195600*, and *PdePrx12* proteins were compared, and the Prx domain was found to be located at 55-295 aa of the PdePrx12 protein sequence (Figure 1C). We have shown the DNA, coding DNA sequence (CDS) and protein sequences of *PdePrx12* in Appendix A.

Using PdePrx12-F and PdePrx12-R as primers and cDNA from NL895 poplar as a template, we amplified the full-length 1065 bp of the *PdePrx12* gene by PCR. The sequence was inserted in a 2301S vector, which was then transferred into *A. tumefaciens* GV3101 to obtain the *A. tumefaciens* overexpression line OE-PdePrx12. Using attb-PdePrx12-F and attb-PdePrx12-R as primers and the full-length sequence of the *PdePrx12* gene as a template, we amplified a *PdePrx12* gene fragment of 266 bp by PCR and inserted it into a pHellsgate4 vector. The vector was subsequently transferred into *A. tumefaciens* GV3101 to obtain a suppressor-expressing *A. tumefaciens* strain RE-PdePrx12.

### 4.2. Expression Characteristics of the PdePrx12 Gene

To investigate the expression of the *PdePrx12* gene in different tissues, RNA was extracted from WT NL895 poplar root, stem, leaf, petiole, and shoot tissues. Setting the expression of *PdePrx12* in root as onefold, its expression was 1.34 ± 0.03-, 5.19 ± 0.39-, 1.66 ± 0.12-, and 1.63 ± 0.09-fold in the stems, leaves, petioles, and shoots, respectively. The expression of *PdePrx12* is highest in leaves (Figure 2A).

To investigate the changes in *PdePrx12* gene expression upon pathogen infection, leaf RNA was extracted after different duration of infection by the 3C/3E pathogens. The relative expression levels of the *PdePrx12* gene in the leaves according to qRT–PCR were 1.03 ± 0.15-, 0.75 ± 0.03-, and 0.48 ± 0.10-fold on Day 0, Day 3, and Day 7, respectively, after 3C infection by the pathogen compared those of with the controls. The relative expression levels of the *PdePrx12* gene in the leaves on Day 0, Day 8, and Day 14 were 1.03 ± 0.03, 0.94 ± 0.02, and 0.59 ± 0.03-fold by qRT–PCR after infection with pathogen 3E compared with those of the controls. Combined with the analysis of the qRT–PCR results, the expression of the *PdePrx12* gene in NL895 poplar was reduced after infection by the pathogen 3C/3E (Figure 2B).

### 4.3. Five OE and Three RE Poplar PdePrx12 Transgenic Lines Were Successfully Generated

To reveal the function of the *PdePrx12* gene, OE-PdePrx12 transgenic lines were successfully obtained by transforming NL895 poplar using *A. tumefaciens* strain OE-PdePrx12; RE-PdePrx12 transgenic lines were successfully transformed and obtained via *A. tumefaciens* strain RE-PdePrx12. In total, there were 21 transgenic lines in both categories, and we randomly selected five OE and three RE lines for further analysis. For the OE transgenic line, we performed positive identification at the DNA, RNA, and protein levels. At the DNA level, it was verified by PCR that the *PdePrx12* gene had been successfully introduced into the genomes of the OE lines OE-PdePrx12-1, OE-PdePrx12-2, OE-PdePrx12-3, OE-PdePrx12-4, and OE-PdePrx12-5 (Figure 3A). At the RNA level, according to the qRT–PCR results, the expression of OE lines OE-PdePrx12-1, OE-PdePrx12-2, OE-PdePrx12-3, OE-PdePrx12-4, and OE-PdePrx12-5 was upregulated 29.04 ± 4.64-, 13.44 ± 0.76-, 177.44 ± 18.01-, 6.61 ± 1.05-, and 15.81 ± 0.98-fold, respectively, compared with that of the controls (Figure 3B). At the protein level, because the OE transgenic line contained a *35S*::*GUS* gene element in its expression vector, we found that, compared with the WT line, the positive OE line was stained blue by GUS staining (Figure 3C). For the RE transgenic lines, we verified the positive identification at both the DNA and RNA levels. At the DNA level, the PCR data showed that fragments of the *PdePrx12* gene were successfully introduced into the genomes of RE lines RE-PdePrx12-10, RE-PdePrx12-15, and RE-PdePrx12-21 (Figure 3D). At the RNA level, the qRT–PCR data showed that the expression of RE lines RE-PdePrx12-10, RE-PdePrx12-15, and RE-PdePrx12-21 was upregulated by 0.61 ± 0.08-, 0.65 ± 0.05-, and 0.13 ± 0.01-fold, respectively, compared with that of the controls (Figure 3E). Through the above validations, we successfully obtained five OE and three RE positive transgenic lines, and these transgenic materials were used for the next steps of our gene functional analysis.

### 4.4. The PdePrx12 Gene Inhibits the Accumulation of H_2_O_2_

To investigate the effect of the *PdePrx12* gene on H_2_O_2_ content, the leaves of WT, OE-PdePrx12-2, OE-PdePrx12-3, OE-PdePrx12-4, and RE-PdePrx12-10, RE-PdePrx12-15, and RE-PdePrx12-21 were selected after 15 days of the plants growing hydroponically to quantify the H_2_O_2_ content. To exclude environmental effects, all the poplar lines used were grown in hydroponic solution under normal conditions. The H_2_O_2_ contents of the WT, OE-PdePrx12-2, OE-PdePrx12-3, OE-PdePrx12-4, RE-PdePrx12-10, RE-PdePrx12-15, and RE-PdePrx12-21 were 0.95 ± 0.01, 0.77 ± 0.01, 0.87 ± 0.00, 0.76 ± 0.01, 1.18 ± 0.03, 1.06 ± 0.00, and 1.36 ± 0.02 mmol/g fresh weight (FW), respectively (Figure 4A). Though the expression of *PdePrx12* in these transgenic lines does not completely match the H_2_O_2_ contents in leaves, we still found that all three OE transgenic lines had significantly lower levels of H_2_O_2_ than did the WT line (*p* < 0.05), whereas all the RE transgenic lines had significantly higher levels of H_2_O_2_ than did the WT line (*p* < 0.05). The inconsistence of the expression of *PdePrx12* with the H_2_O_2_ contents in leaves can be attributed to the uncorrected gene expression analyses of some transgenic lines.

DAB is a color indicator for H_2_O_2_, and leaves of WT, OE, and RE tissue culture plants were collected for DAB staining. The WT leaves were brown, the leaves infected with the OE line were yellow, and the leaves infected with RE line were dark brown (Figure 4B). These results also indicated that the OE transgenic line had a lower H_2_O_2_ content than the WT line, whereas the RE transgenic line had a significantly higher H_2_O_2_ content than both WT lines. Therefore, based on the above data, it can be concluded that the expression of the *PdePrx12* gene can inhibit the accumulation of H_2_O_2_.

### 4.5. The PdePrx12 Gene Negatively Regulates Disease Resistance in Poplar

The above selected transgenic lines were grown in soil for 2 months and inoculated with the 3C/3E pathogens. After 10 days of dark incubation, it was observed that the leaves of the OE line were infected with pathogen 3C up to the margins (Figure 4C), and compared to the WT, all the OE lines were extensively infected with pathogen 3C, whereas all the RE lines were less infested. Measurements of the absolute areas of individual spot infection development were performed. The mean infected areas of the WT, OE-PdePrx12-1, OE-PdePrx12-2, OE-PdePrx12-3, OE-PdePrx12-4, OE-PdePrx12-5, RE-PdePrx12-10, RE-PdePrx12-15, and RE-PdePrx12-21 were 1.65 ± 0.30, 3.66 ± 0.15, 4.83 ± 0.39, 2.80 ± 0.47, 3.92 ± 0.68, 2.58 ± 0.72, 0.98 ± 0.53, 0.72 ± 0.38, and 0.49 ± 0.22 cm^2^, respectively. After 15 days of dark incubation, it was observed that the leaves of the OE line were infected with pathogen 3E up to the margins (Figure 4D). Like the results concerning pathogen 3C, compared with the WT, all the OE lines were extensively infected by pathogen 3E, whereas the RE lines were all less infested. Measurements of the absolute areas of individual spot infection development were performed using Image-Pro Plus image processing software. The mean infected areas of the WT, OE-PdePrx12-1, OE-PdePrx12-2, OE-PdePrx12-3, OE-PdePrx12-4, OE-PdePrx12-5, RE-PdePrx12-10, RE-PdePrx12-15, and RE-PdePrx12-21 were 4.77 ± 0.57, 10.66 ± 0.90, 8.68 ± 0.12, 5.85 ± 0.82, 6.55 ± 0.35, 6.65 ± 0.12, 2.59 ± 0.65, 1.48 ± 0.08, and 1.65 ± 0.21 cm^2^, respectively. The experimental results showed that under the infection of pathogen 3C/3E, the leaf infection area of the OE line was larger than that of the WT NL895 poplar, and the leaf infection area of the RE line was smaller than that of the WT NL895 poplar. By ANOVA and multiple comparison analysis, the leaf infection area of OE-PdePrx12-1, OE-PdePrx12-2, and OE-PdePrx12-4 was significantly larger than that of WT NL895 poplar at the time of 3C infection by the pathogen; additionally, that of the other lines was not significantly different from the results of WT NL895 poplar, but there were some differences compared with the data of WT NL895 poplar (Figure 4E). At the time of pathogen 3E infection, the leaf infected area for all lines was significantly different from that of WT NL895 poplar except for the OE-PdePrx12-3 line, whose data were not significantly different from those of WT NL895 poplar (Figure 4F). Thus, this suggests that overexpression of the *PdePrx12* gene in NL895 poplar decreased the resistance of NL895 poplar to 3C/3E pathogens, whereas inhibition of *PdePrx12* gene expression increased the resistance of NL895 poplar to 3C/3E pathogens. According to the analysis of the above experimental results, the *PdePrx12* gene negatively regulates disease resistance in poplar.

## 5. Discussion

H_2_O_2_ is important for pathogen defense and signal transduction, and H_2_O_2_ accumulates in a variety of plant disease systems, where it can directly kill pathogenic microorganisms or inhibit spore germination. When they infect plants, pathogens secrete enzymes that degrade the cell wall, and H_2_O_2_ increases plant resistance to these enzymes [29]. Whereas H_2_O_2_, as a signaling molecule, can be produced rapidly after pathogen infection, the accumulation of H_2_O_2_ in plants and the subsequent hypersensitivity are considered markers of plant resistance to pathogen invasion [30,31]. Studies have shown that Prxs are important oxidative enzymes in plants [32] and are key enzymes in the plant defense system. They play key roles in various metabolic pathways, such as reactive oxygen species metabolic pathways [33,34]. In the present study, we hypothesized the disease resistance mechanism of the *PdePrx12* gene and summarized the regulatory model for its involvement in the disease resistance response. According to this model, *PdePrx12* gene expression is repressed when plants are infected with pathogens, resulting in increased H_2_O_2_ levels in poplar and disease resistance. Based on this, we can speculate that poplar responds actively to pathogen invasion through changes in *PdePrx12* gene expression, which in turn leads to enhanced disease resistance.

As far as we know, there are only a few studies available that focus on revealing the gene function of PRX in trees. Meanwhile, according to previous studies, PRX is widely involved in multiple biological processes. One important aspect is that PRX participates in the oxidation process of the growth hormone indoleacetic acid IAA [35]. IAA plays an important role in regulating plant cell elongation, apical dominance, rooting, dormancy, and vernalization. Therefore, research on the *PdePrx12* gene can be centered on auxin (IAA) to study aspects such as adventitious root development in poplar. It was found that tomato cells in the presence of BcGS1(a glycoprotein from *Botrytis cinerea*) promoted the production of hydrogen peroxide, which increased the activity of peroxidase to further spontaneously couple the production of monophenolic radicals to form lignin aggregates; the experiment illustrated the ability of peroxidase to prevent further cell damage through the control of lignin synthesis [36], so the *PdePrx12* gene can also be studied in the context of lignin for wood development, disease and stress resistance, etc.

## Figures and Tables

**Figure 1 jof-09-00410-f001:**
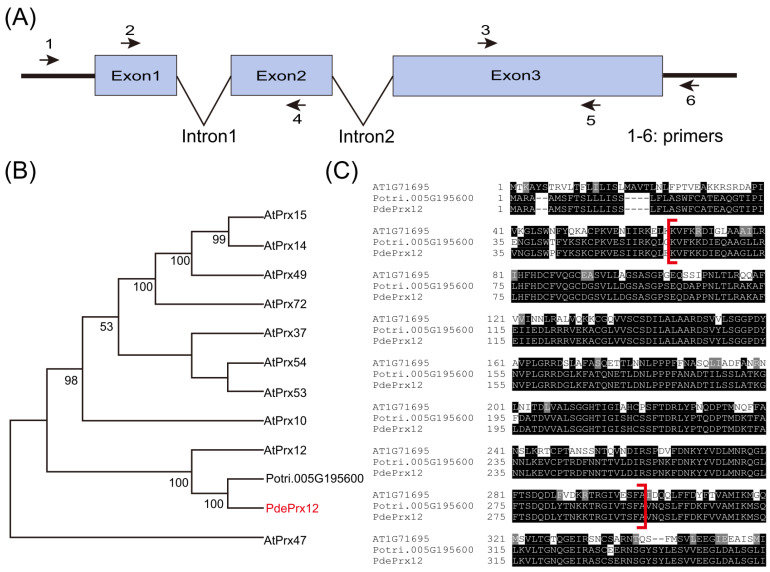
Characteristics of the *PdePrx12* gene. (**A**) Structure of the PdePrx12 gene and primer positions designed according to Potri.005G195600. Numbers 1, 2, 3, 4, 5, and 6 represent the relative positions of primers PdePrx12-F, attb-PdePrx12-F, PdePrx12-qpc-F, attb-PdePrx12-R, PdePrx12-qpc-R, and PdePrx12-R, respectively. (**B**) Phylogenetic analysis of the PdePrx12 gene and similar proteins. For phylogenetic analysis of PdePrx12 and similar proteins, the 10 genes most similar to PdePrx12 genes in *Arabidopsis thaliana* and the Potri.005G195600 gene most similar to PdePrx12 in *P.trichocarpa* were selected to construct phylogenetic trees. The PdePrx12 gene is indicated by a red arrow. (**C**) Protein sequence alignment of AT1G71695 (AtPrx12), Potri.005G195600 and PdePrx12 (the red brackets indicate the Prx domain).

**Figure 2 jof-09-00410-f002:**
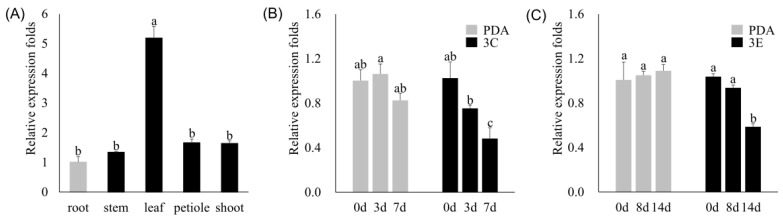
Expression characteristics of the *PdePrx12* gene. (**A**) Expression of the *PdePrx12* gene in the roots, stems, leaves, petioles, and terminal buds of wild-type poplar NL895. The relative expression of *PdePrx12* in the roots was set to onefold (*p* = 0.05). (**B**) Expression of *PdePrx12* on Day 0, Day 3, and Day 7 when 3C infects WT. The relative expression level of *PdePrx12* in Day 0 WT leaves was set to onefold (*p* = 0.05). (**C**) Expression of *PdePrx12* on Day 0, Day 8, and Day 14 when 3E infects WT. The relative expression level of *PdePrx12* in Day 0 WT leaves was set to onefold (*p* = 0.05). Different letters (a, b, c and ab) above bars indicated significant difference at *p* = 0.05.

**Figure 3 jof-09-00410-f003:**
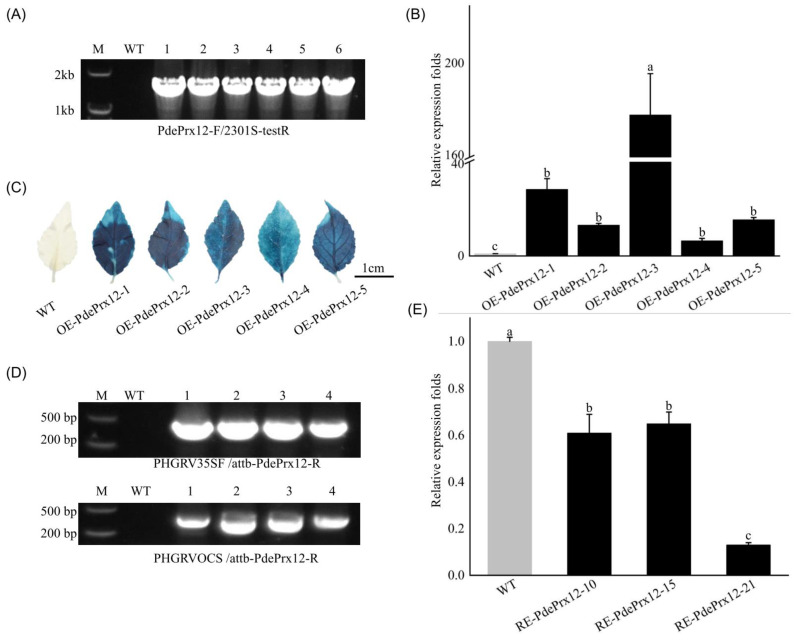
Generation of transgenic poplar lines. (**A**) The presence of the overproducing construct in transgenic poplar. Primers PdePrx12-F/2301S-testR were used for PCR of the OE line, where M represents the marker; WT represents WT NL895 poplar; 1 represents the 2301S-PdePrx12 plasmid; and 2, 3, 4, 5, and 6 are five positive lines of OE-PdePrx12. (**B**) Positive identification of RNA level in *PdePrx12* transgenic lines. Relative expression of *PdePrx12* in the WT and OE lines. The relative expression level of *PdePrx12* in WT was set to onefold. Bars indicate standard deviation (*p* = 0.05). (**C**) Comparison of GUS staining between OE lines and WT lines (bar = 1 cm). (**D**) The presence of the gene silencing construct in transgenic poplar. The DNA level of RE-PdePrx12 transgenic lines was positively identified. The primers PHGRV35SF/attb-PdePrx12-R and PHGRVOCS/attb-PdePrx12-R were used for PCR in the RE lines, where M represents the marker; WT represents wild-type NL895 poplar; 1 represents the PHGRV-PdePrx12 plasmid; and 2, 3 and 4 are the three positive lines of RE-PdePrx12. (**E**) Positive identification of RNA level in *PdePrx12* transgenic lines. Relative expression of *PdePrx12* in the WT and RE lines. The relative expression level of *PdePrx12* in WT was set to one-fold. Bars indicate standard deviation. Different letters (a, b and c) above bars indicated significant difference at *p* = 0.05.

**Figure 4 jof-09-00410-f004:**
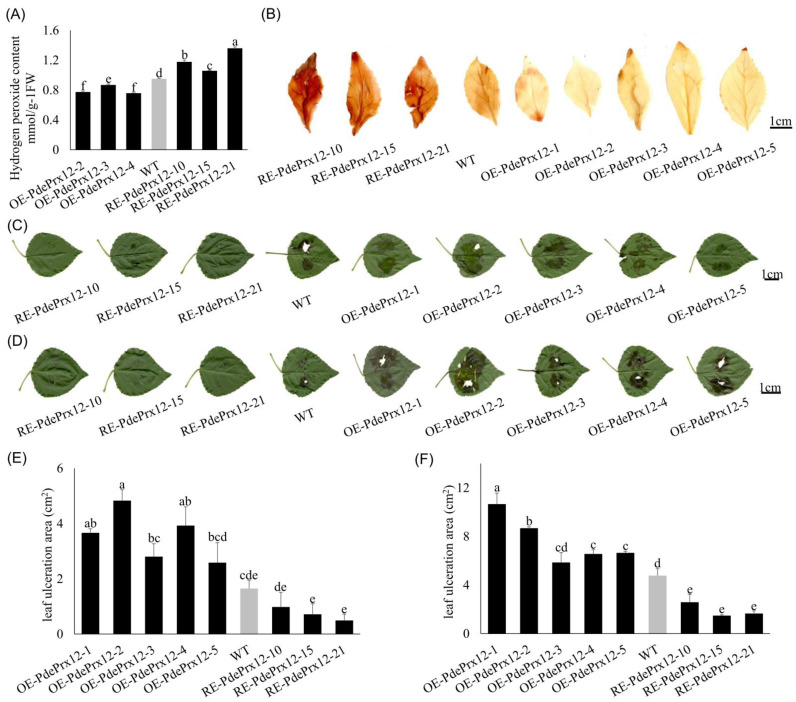
Phenotypes of the *PdePrx12* transgenic lines. (**A**) Determination of H_2_O_2_ content in OE/RE-PdePrx12 and WT lines (mmol/g FW). Determination of H_2_O_2_ content in OE/RE-PdePrx12 and WT lines. The content of H_2_O_2_ in WT was set to 1. The letters in the figure indicate differences based on multiple comparisons (*p* = 0.05). (**B**) DAB staining of OE/RE-PdePrx12 and WT leaves (bar = 1 cm). (**C**) Phenotypic observation of OE/RE-PdePrx12 and WT under the infection of pathogen 3C on the 7th day (bar = 1 cm). (**D**). Phenotypic observation of OE/RE-PdePrx12 and WT under the infection of pathogen 3E on the 14th day (bar = 1 cm). (**E**) The mean values of the infected area of leaves of OE, RE and WT lines by pathogen 3C were statistically analyzed by Image-Pro Plus image processing software. The letters in the graphs indicate differences based on multiple comparisons (*p* = 0.05). (**F**) The mean values of the infected area of leaves of OE, RE, and WT lines by pathogen 3E were statistically analyzed by Image-Pro Plus image processing software. The letters in the graphs indicate differences based on multiple comparisons (*p* = 0.05).

## Data Availability

The data presented in this study are available in Appendix A.

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
