# Peer review of "Uncovering the Role of PdePrx12 Peroxidase in Enhancing Disease Resistance in Poplar Trees"

_jof, 2023, doi:10.3390/jof9040410_

Round 1
Reviewer 1 Report
The paper should be significantly improved before publication.
One of the main points is that, at least in the case of overexpression of the PdePrx12 there is no direct evidence about protein expression. The authors proved the presence of the gene and gene transcripts in transgenic plants. Although they make GUS staining in overexpressing plants but the 35S::GUS protein activity is not prove the translation of the PdePrx12 (as it not a gene fusion construct). Direct protein detection, peroxidase activity or peroxidase isoenzyme activity detection should demonstrate the protein level changes.
Other comments:
Abstract:
-line 44 “spectrophotometric analysis” instead of “via spectrophotometry examination”.
-line 45 “RE line had a increased H2O2 content” instead “RE line had a reduced H2O2 content”.
Introduction:
-The functions of the peroxidase (e.g. H2O2 production) could be supplemented with more details
and more cited literature should incorporate into the text.
-line 76: “in many plant biological activities” instead “in all plant biological activities”.
-form of the Arabidopsis writing should be unified (italic?)
-line 102-109. This section should be moved after the section on peroxidase.
-line 104. In many cases, a semicolon is used. It would be better to use new sentences in these cases.
Materials and Methods:
-line 128 and 133: The text should include references of the media.
-line 131-132: The origin of the pathogens must be indicated (references).
-line 138: details the inoculation method should incorporate into the text.
-line 138: “PDA” instead “PAD”.
-line 148: the exact name of the Kit should be wrote into the text.
-line 154: what is meant by biological replicates should be clarified.
-line 155: gene identification number of the actin should be indicated in the text.
-line 161: the name/type of the Taq polymerase should be indicated in the text (proofreading?).
-line 161: the sequence (GenBank accession number) should be indicated in the text.
-line 162: references of the plasmid vectors should be indicated in the text.
-line 201: “Phenotypic observations of the symptoms” or similar instead” Phenotypic observations”.
-line 203: “histopathogen bottles” is probably a wrong name.
-line 204-206: more details of plant incubation conditions should be indicated in the text.
-line 208: how did they determined 4th-10th noncurled normal leaves?
-line 208: it is not obvious that they worked with detached leaves.
-line 211: what does the “same size” mean?
-line 221: the internet address of the Arabidopsis database website should be clarified.
Results
-line 234: the name of the database of the gene Potri.005G195600 should be indicated in the text (Phytozome?).
-line 312-318. The strength of the DAB staining reaction is depends of the reaction between peroxidase and H2O2. As they manipulate the amount of peroxidase the method is not suitable to detect the amount of H2O2.
Discussion
-line 359: not sure if correct reference were cited.
-line 360: The term "allergic" is not used for plant reactions.
-line 372-377: long sentence.
-line 382: what does Bc GS1 mean?
Figure and Figure legends:
Figure 1 C
-PdePrx12 instead PdePrx25
-how did they found the PRX domain (method)
Figure 2 A B C
-0d, 3d, 7d should place under the x axis.
-the figures do not show the results of significances.
-the meaning of the error bars should be indicated.
-line 494-496 hard to understand sentence that needs to be rewritten.
Figure 3
-line 501 “Identification of the presence of the PdePrx12 DNA” or similar instead of “Positive identification of the DNA level of PdePrx12 transgenic lines.” There are several similar ones in the text.
Figure 3 A
-the marker lines are not visible on the figure
Figure 3 B and E
-the figures do not show the results of significances.
-the meaning of the error bars should be indicated.
Figure 4. C D E F
-The day of sampling must be entered in the text.
Figure 5.
-the figure does not provide new information, so it can be omitted.
Table S1.
- pdePrx12-qpc-F and pdePrx12-qpc-R are not reference gene primers.
Reviewer 2 Report
1. Please explain the reasons why the authors selected only five OE and three RE transgenic lines for further analysis.
2. How were the H2O2 contents of the OE-PdePrx12-1 and OE-PdePrx12-5? Were they also lower than that of WT?
3. Expression of the PdePrx12 gene at the DNA and RNA levels was highest in the OE-PdePrx12-3 line. Why was the level of H2O2 in the line was also highest compared to the other two OE lines?
4. Please explain the reasons why the OE-PdePrx12-3 line had the highest H2O2 level among the OE lines but was not reflected in the resistance level of the line against both 3C and 3E.

Round 2
Reviewer 1 Report
The manuscript still needs to be improved before publication.
I still miss the direct evidence about peroxidase protein expression/activity changes in transgenic lines (direct protein detection, peroxidase activity or peroxidase isoenzyme activity detection (e.g. PAGE) should demonstrate the protein level changes). I leave it to the editor's decision whether to accept the manuscript without these tests.
There are some shortcomings that I mentioned before, but they were not fixed in the new version available for me, even though the answers promised to fix them:
(the numbering of the lines corresponds to the original manuscript)
Abstract:
-line 44 “spectrophotometric analysis” instead of “via spectrophotometry examination”.
-line 45 “RE line had a increased H2O2 content” instead “RE line had a reduced H2O2 content”.
Introduction:
-The functions of the peroxidase (e.g. H2O2 production) could be supplemented with more details and more cited literature should incorporate into the text.
-line 76: “in many plant biological activities” instead “in all plant biological activities”.
-form of the Arabidopsis writing should be unified (italic?)
-line 104. In many cases, a semicolon is used. It would be better to use new sentences in these cases.
Materials and Methods:
-line 131-132: The origin of the pathogens must be indicated (references).
-line 138: details the inoculation method should incorporate into the text.
-line 154: what is meant by biological replicates should be clarified in the text not only in the answers.
-line 161: the name/type of the Taq polymerase should be indicated in the text (proofreading?).
-line 161: the sequence (GenBank accession number) should be indicated in the text.
-line 162: references of the plasmid vectors should be indicated in the text.
-line 201: “Phenotypic observations of the symptoms” or similar instead” Phenotypic observations”.
-line 204-206: more details of plant incubation conditions should be indicated in the text.
-line 208: it is not obvious that they worked with detached leaves. It should be indicated into the text
-line 221: the internet address of the Arabidopsis database website should be clarified. Which database was used?
Results
--line 312-318. Fig 4B. I still think that as the amount of peroxidase was manipulated in the plants the DAB staining method is not suitable to detect the amount of H2O2. This is because the strength of the DAB staining reaction is depends on the reaction between peroxidase and H2O2. It can only be stated that the DAB staining show changes in leaves, but it is not possible to know to what extent H2O2 and peroxidase activity are involved in the formation of the staining intensity.
Discussion
-line 382-385 (new version of MS) “Clarify the function of….”. the sentence should be rewritten.
Figure and Figure legends:
Figure 2 C. the sentence should be rewritten.
Figure 3 A and D. They did not show the presence of the PdePrx12 gene, but the presence of the overproducing and gene silencing construct in transgenic plants. The sentences should be rewritten.
Author Response
Dear Reviewer
Thank you for your effort on our manuscript. Your comprehensive comments and suggestions were very helpful in improving our manuscript. We have addressed your comments as much as we could and we hope this version could meet the requirements of this journal.
- I still miss the direct evidence about peroxidase protein expression/activity changes in transgenic lines (direct protein detection, peroxidase activity or peroxidase isoenzyme activity detection (e.g. PAGE) should demonstrate the protein level changes). I leave it to the editor's decision whether to accept the manuscript without these tests.
Response: Thanks for your comments, however, I think it is not a question. The transgenic technique in poplar is well developed. Almost all similar studies do not perform protein expression assay for poplar transgenic plants.
- Abstract:
-line 44 “spectrophotometric analysis” instead of “via spectrophotometry examination”.
-line 45 “RE line had a increased H2O2 content” instead “RE line had a reduced H2O2 content”.
Response: These points were corrected.
- Introduction:
-The functions of the peroxidase (e.g. H2O2 production) could be supplemented with more details and more cited literature should incorporate into the text.
Response: We already added more information for this point.
-line 76: “in many plant biological activities” instead “in all plant biological activities”.
-form of the Arabidopsis writing should be unified (italic?)
-line 104. In many cases, a semicolon is used. It would be better to use new sentences in these cases.
Response: We have corrected these points.
- Materials and Methods:
-line 131-132: The origin of the pathogens must be indicated (references).
Response: We have corrected it. And we cite the references in their first appearance.
- -line 138: details the inoculation method should incorporate into the text.
Response: We describe it in detail in the text.
- -line 154: what is meant by biological replicates should be clarified in the text not only in the answers.
Response: We describe it in the text.
-line 161: the name/type of the Taq polymerase should be indicated in the text (proofreading?).
Response: We described it in the text.
-line 161: the sequence (GenBank accession number) should be indicated in the text.
Response: PdePrx12 was cloned from poplar line NL895 and its alleles in Populus trichocarpa v3.0 is Potri.005G195600. We provided its sequence in the supplementary file.
-line 162: references of the plasmid vectors should be indicated in the text.
-line 201: “Phenotypic observations of the symptoms” or similar instead” Phenotypic observations”.
Response: We have corrected these points.
-line 204-206: more details of plant incubation conditions should be indicated in the text.
Response: We have added the conditions for hydroponics and relevant references to the text.
-line 208: it is not obvious that they worked with detached leaves. It should be indicated into the text.
Response: We have corrected it in the last modification.
-line 221: the internet address of the Arabidopsis database website should be clarified. Whic.h database was used?
Response: We have corrected these points.
- Results
--line 312-318. Fig 4B. I still think that as the amount of peroxidase was manipulated in the plants the DAB staining method is not suitable to detect the amount of H2O2. This is because the strength of the DAB staining reaction is depends on the reaction between peroxidase and H2O2. It can only be stated that the DAB staining show changes in leaves, but it is not possible to know to what extent H2O2 and peroxidase activity are involved in the formation of the staining intensity.
Response: Thanks for your comments. However, we already provided the exact number of H2O2 content in Fig. 4A. DAB staining is a supplementary data for Fig. 4A. Thus, we think it might be suitable to provide DAB staining data here.
- Discussion
-line 382-385 (new version of MS) “Clarify the function of….”. the sentence should be rewritten.
Response: We reorganized this paragraph.
- Figure and Figure legends:
Figure 2 C. the sentence should be rewritten.
Response: We reorganized this paragraph.
Figure 3 A and D. They did not show the presence of the PdePrx12 gene, but the presence of the overproducing and gene silencing construct in transgenic plants. The sentences should be rewritten.
Response: We have corrected these points.
